# Evaluation of Peri-Operative Outcomes after Prostatic Urethral Lift with Emphasis on Urodynamic Changes, Symptom Improvement and Sexual Function

**DOI:** 10.3390/diagnostics14192110

**Published:** 2024-09-24

**Authors:** Riccardo Lombardo, Valerio Santarelli, Beatrice Turchi, Giuseppe Santoro, Alessandro Guercio, Antonio Franco, Silvia Secco, Paolo Dell’Oglio, Antonio Galfano, Alberto Olivero, Antonio Luigi Pastore, Yazan Al Salhi, Andrea Fuschi, Antonio Nacchia, Giorgia Tema, Alessandra Fegiz, Ferdinando Fusco, Riccardo Cini, Antonio Cicione, Andrea Tubaro, Cosimo De Nunzio

**Affiliations:** 1Department of Urology, Sapienza Università di Roma, 00191 Rome, Italy; valerio.santarelli@uniroma1.it (V.S.); beatrice.turchi@uniroma1.it (B.T.); giuseppe.santoro@uniroma1.it (G.S.); alessandro.guercio@uniroma1.it (A.G.); antonio.franco@uniroma1.it (A.F.); yazan.alsalhi@uniroma1.it (Y.A.S.); andrea.fuschi@uniroma1.it (A.F.); antonio.nacchia@uniroma1.it (A.N.); giorgia.tema@uniroma1.it (G.T.); alessandra.fegiz@uniroma1.it (A.F.); ferdinando.fusco@gmail.com (F.F.); riccardo.cini@uniroma1.it (R.C.); cosimo.denunzio@uniroma1.it (C.D.N.); 2Department of Urology, Ospedale Niguarda, 20162 Milan, Italy; silvia.secco@ospedaleniguarda.it (S.S.); paolo.delloglio@gmail.com (P.D.);

**Keywords:** urodynamic, minimally invasive, benign prostatic hyperplasia

## Abstract

**Background and Aims:** The aim of our study is to evaluate the possible urodynamic effect of prostatic urethral lift (PUL) in patients with lower urinary tract symptoms due to benign prostatic hyperplasia. **Methods**: A consecutive series of patients undergoing PUL placement were consecutively enrolled in two centers. Inclusion criteria: ≥50 years of age, benign prostatic obstruction (BPO), international prostate symptom score (IPSS) ≥ 13, prostate volume ≤ 60 mL, and no middle prostate lobe. All patients were evaluated using a detailed clinical history, a validated questionnaire, flexible cystoscopy, and pressure flow studies (PFS) at baseline. PFS were performed at 6 months to evaluate the urodynamic effect of PUL. **Results**: Overall, 20 patients with a median age of 63 were enrolled. At six months, statistically significant improvements in terms of median Qmax (11.5 vs. 8.5; *p* < 0.05) and median IPSS (16 vs. 10.5; *p* < 0.05) were recorded, and sexual function was maintained. All urodynamic parameters improved at 6 months, and significance was reached for all values except for PdetQmax. Finally, Schäfer’s class improved from a median of III to a median of II. More specifically, 16/20 presented an improvement in the Schäfer class, and 12/20 patients presented a BOOI < 20 at 6 months. **Conclusions:** PUL represents an effective treatment in patients with LUTS due to BPH and improves bladder outlet obstruction without any effect on sexual function.

## 1. Introduction

Bladder outlet obstruction (BOO) caused by benign prostatic hyperplasia (BPH) is one of the main causes of Lower Urinary tract symptoms (LUTS) in adult men [1]. In patients who fail medical treatment, endoscopic procedures and simple prostatectomy, depending on prostate size, represent the gold standard to relieve BOO. However, recently the prostatic urethral lift (PUL) has been introduced in the European Association of Urology guidelines as a minimally invasive alternative to more invasive procedures for the treatment of BOO due to BPH.

Transurethral resection of the prostate (TURP) is still considered by several authors the gold standard for the treatment of BOO caused by BPH [1]. However, most of the patients present retrograde ejaculation after surgery, causing severe discomfort, particularly in young patients. To overcome this limitation, several ejaculation-sparing techniques have recently been introduced. For instance, water thermal vapor therapy, iTind, trans-perineal laser ablation of the prostate, prostatic artery embolization, and Aquablation. Some of these techniques are still to be considered investigational and cannot be compared in terms of functional outcomes to the classic interventions, although their invasiveness is more favorable. As for the time being, their place is between medical treatment and definitive surgical intervention [2,3,4,5].

PUL achieves quantifiable improvements in terms of symptom release (IPSS + 9.40 points, *p* < 0.001), QoL (+1.99 points, *p* < 0.001), and uroflowmetry (Qmax + 3.39 mL/s, *p* < 0.001), without major adverse events and with good preservation of ejaculatory ad sexual function. The PUL procedure involves the delivery of implants that retract the obstructing prostate lobes under cystoscopy visualization. Thus it is claimed to be designed to relieve BOO [6]. Despite that, while studies have demonstrated that in selected populations, symptomatic improvement after PUL is comparable to that obtained by TURP, flowmetry parameters only improve moderately, and this discrepancy is still poorly understood [7]. 

The role of PUL in relieving BOO has been poorly evaluated, and only one study is available reporting urodynamic data. According to Muller et al., PUL presented a moderate effect on urodynamic parameters, and the Schäfer class improved by a median of one class. The study by Muller is the only study on the subject available in the literature. However, it takes more than one study to prove a hypothesis. With this knowledge in mind, the aim of our study is to evaluate whether PUL treatment is associated with an improvement in urodynamic parameters. Secondary objectives are the evaluation of the degree of symptomatic response, preservation of sexual function, and median return to normal activities. 

## 2. Materials and Methods

After obtaining ethical approval, all patients undergoing PUL who met the inclusion criteria during the enrollment period were asked to participate in the study. All patients signed an informed consent, and the study was conducted in accordance with the Helsinki Declaration. Inclusion criteria were the following: male sex, age ≥ 50 years, prostate volume (PV) ≤ 60 cc, IPSS score ≥ 13, and no middle prostate lobe. Exclusion criteria included patients with Qmax < 5 mL/s; PVR > 150 cc, middle prostatic lobe, urethral strictures, previous prostate or bladder surgery, and Schäfer class < 2.

At baseline, all patients were preoperatively evaluated with a thorough clinical and pharmacological history. LUTS were evaluated using the International Prostatic Symptom Score (IPSS), erectile function with the International Index for erectile function (IIEF-5), and ejaculatory function with the male sexual health (MSHQ) questionnaires. 

All patients underwent flexible cystoscopy before the procedure to exclude urethral stenosis and to evaluate the presence of a median lobe. 

A transurethral pressure-flow study was performed in all patients according to the International Continence Society recommendations at baseline and 6 months after the procedure. PFS parameters included maximum flow rate (Qmax), detrusor pressure at a maximum flow rate (PdetQmax), opening detrusor pressure (PdetOp), and detrusor pressure at minimum flow (PdetVoidMin). PFS parameters were plotted on the 1993 version of the Schäfer nomogram to obtain the Schäfer class. PFSs were performed blind of any other clinical parameter. BOO was defined as the presence of a Schäfer Class ≥ 3.

The urodynamic testing was conducted using a multichannel system. Fluid-filled lines and an external transducer were utilized during the investigations. For bladder pressure measurement and filling, an 8F transurethral dual-lumen catheter was employed. Additionally, an 8F rectal balloon catheter was used to record abdominal pressure, enabling the determination of detrusor pressure. Midstream urinalysis, including urinary sediment and culture performed before UDS, yielded negative results at the time of the urodynamic evaluation. All patients provided informed consent before the test. Prior to cystometry, the bladder was emptied through the lumen of the transurethral catheter, and the transducers were balanced to atmospheric pressure. The bladder was then filled with sterile water at 20 °C at a rate of 50 mL/min. The maximal flow, detrusor pressure at maximal flow, and minimal urethral opening pressure were plotted on the 1993 version of the Schäfer nomogram to determine the linear passive urethral resistance relation and the Schäfer obstruction class.

All patients underwent the PUL procedure under spinal anesthesia. The Prostatic Urethral Lift system (Prostatic Urethral Lift™; Neotract Inc., Pleasanton, CA, USA) utilizes a 2.9 mm 0° lens and a 20F cystoscopy sheath with a custom bridge. This single-use delivery device consists of three components: a nitinol capsular tab for the prostate capsule, a stainless-steel urethral end piece, and a permanent polyethylene terephthalate suture that provides tension. Under the endoscopic transurethral vision, after mechanically compressing the prostatic lobe, the nitinol anchor is passed through the parenchyma and anchored to the prostate capsule. The implant is then released using precise triggers. The devices were laterally positioned, 1.5 cm distal to the bladder neck, and released at an angle of 20–30° to the right and left of the 12 o’clock position. The goal of the technique was to maximally enlarge the prostatic urethra, with additional devices being deployed as necessary to achieve the desired result. The mean opening of the anterior chamber of the prostatic urethra was confirmed cystoscopically. All procedures were performed by a single senior surgeon (CDN) under spinal anesthesia. A urethral catheter was left in place and removed on postoperative day 1 before discharge (Figure 1).

Operative time, ASA score, and any intraoperative complication that occurred were recorded. The number of implants needed for each patient was also reported. 

In the postoperative setting, the duration of hospitalization and catheterization time were evaluated. As well, patients were evaluated with recovery time in terms of time to normal activity, time to work, time to physical activity, and time to sexual activity expressed in days. 

Follow-up visits and diagnostic workups were carried out as follows: Patients completed the IPSS, IIEF-5, and MSHQ questionnaires at 6 months. PFS were also repeated at 6 months to evaluate the functional improvement obtained with the PUL procedure at a mid-term follow-up. 

Statistical analysis was carried out using SPSS (V25, IBM, Armonk, NY, USA). Continuous data were expressed in terms of median and quartiles. The difference between the baseline and follow-up median was calculated, and significance (*p* < 0.05) was analyzed using the Wilcoxon signed-rank test. Data are presented as Median (Interquartile range). 

## 3. Results

Overall, a total of 20 patients with a median age of 63 years were enrolled in the study. Patients’ characteristics are listed in Table 1. 

All PUL procedures were carried out in 2023. The more frequent number of anchors implanted was 2 (45%, 9/20), followed by 4 (40%, 8/20). 3, 5, and 6 anchors were implanted in 1 patient each. Median operative time and ASA score were, respectively, 15 (12–20) min and 2 (1–3). No major intraoperative complications were recorded (Figure 2). 

Overall, there was a significant improvement in terms of symptom relief. Median IPSS after 6 months of treatment was 10.5 (7.25–11.75), with a 5.5-point reduction (34.4%, *p* < 0.001). Improvement was registered in both the storage (sIPSS) and voiding (vIPSS) domains of the IPSS questionnaire. vIPSS decreased by 47.37% (4.5 points), while sIPSS decreased by 38.5% (2.5 points). Regarding sexual function, median IIEF-5 at 6 months remained stable (23 vs. 22; *p* = 0.255). MSHQ(Q1 + Q2 + Q3) at 6 months was 12.5, compared to 10.0 preoperatively (*p* = 0.045), suggesting an improvement in ejaculatory function. All patients stopped pharmacological treatment after surgery (Table 2). 

All urodynamic parameters improved at 6 months, and significance was reached for all values except for PdetQmax. Median UDN-Qmax increased by 37.5% (3/8), while median BCI and BOOI improved by 5% (5/95) and 54% (21/39). Median Schäfer class at 6 months decreased to II (*p* = 0.006), and 16/20 improved their Schäfer class (Table 3). Overall, 12/20 patients (60%) presented a BOOI < 20 after PUL positioning. 

Patients returned to their normal activity after a median of 8 (5–11.5) days, and after a median of 10.5 (7–14) days, they were able to go back to work. Median time for physical activities and sexual activities were, respectively, 20 (15–24.75) and 22.5 (15–30) days. 

## 4. Discussion

PUL is a minimally invasive technique for the treatment of male LUTS caused by BPH. It is intended for men who do not respond to medical therapy or who are not compliant with the idea of chronic medication and its adverse events (AEs), but at the same time are not willing to undergo more invasive procedures like TURP that hold a higher chance of immediate or long-term complications. The present study and Muller study are the only available research in the literature evaluating the urodynamic effects of PUL. In our study, a six-month improvement in both subjective (IPSS score) and objective (Qmax) outcomes was observed. Median IPSS after 6 months of treatment was 10.5 (7.25–11.75), with a 5.5 point reduction (34.3%, *p* < 0.001), and these results are in line with those shown in the previous literature [8,9]. Not surprisingly, the relief in voiding symptoms was superior to that in storage symptoms. Nonetheless, a decrease in tension in the bladder neck can cause a reduction in urge sensation, which in turn reduces storage LUTS [10]. The symptomatic improvement provided by the PUL procedure is superior to that reported for medical therapy or placebo, which both improve IPSS at 12 months by 3.5–7.5 points but not to that observed for TURP [11]. IPSS improvement after transurethral resection is reported to be >15 points, and a recent Cochrane review has shown that PUL may result in substantially lower improvement in IPSS score (MD 6.10, 95% CI 2.16–10.04) compared to TURP [12,13].

The objective improvement after PUL, measured with the modification of the urodynamic parameters defining obstruction after the procedure, was the primary objective of our study. The superolateral traction of the lateral prostatic lobe provided by the PUL procedure resulted in a 35.2% (3/8.5) median Qmax improvement in our series. Our results are in line with those reported by previous authors despite the high heterogeneity of Qmax outcomes reported. This high heterogeneity implies the importance of adequate patient selection and surgical experience and suggests that, for now, improvements in Qmax after PUL should be considered with caution. The functional improvements observed in the current study appear non-inferior when compared with medical therapy [14], which shows a 27% improvement in Qmax after 12 months, dropping to 8–11% after 2 years of follow-up [15,16].

In the present study, we meticulously detailed the urodynamic procedures to ensure our data would be reproducible. Unfortunately, Miller et al. did not specify how their urodynamic studies were performed, which is a limitation of their study. Additionally, we carefully evaluated sexual function before and after the procedure, as this is often a primary goal of Prostatic Urethral Lift surgery. Consistent with existing data on sexual function following the Prostatic Urethral Lift procedure, all our patients maintained their sexual function post-intervention. Furthermore, in our study, bladder outlet obstruction (BOO) was assessed using the Bladder Outlet Obstruction Index (BOOI), which is the gold standard for BOO evaluation. In contrast, Miller et al. only considered the Schäfer classification. According to our results, 12/60 (60%) patients reached a BOOI < 20 after Prostatic Urethral Lift positioning. Lastly, our study demonstrates the urodynamic effect of PUL at six months, whereas Miller et al. evaluated patients at three months [17,18,19].

The greatest advantage of PUL, contrary to more invasive surgical therapies, is observed in sexual outcomes. The literature reports increased or maintained sexual health in patients who underwent the PUL procedure. TURP, or photoselective Vaporization of the Prostate (PVP), is associated with high rates of ejaculatory dysfunction (15–63%), which often discourages patients from undergoing surgical treatment. As well, the impact of medical treatment on sexual health could be associated with the loss of libido or antegrade ejaculation [17]. In our study, after 6 months, there was a significant increase in sexual health measured with the IIEF-5 questionnaire and an improvement in ejaculatory function.

In the past years, the introduction of minimally invasive techniques has enriched the surgical armamentarium to relieve BOO due to BPH. PUL is probably a better fit for younger patients with the desire to preserve anterograde ejaculation and avoid the risk of long-term complications and also for older patients desiring to avoid the risks of more invasive procedures under general anesthesia. Likewise, avoiding the physical and psychological effects of long-term medication use could be another reason to opt for MITs. The growing literature on the subject suggests MITS may have a role in medical and definitive treatment. Physicians should always keep in mind a patient-centered approach in BPH management. Patient symptoms, social context, preferences, and expectations should be fundamental in the counseling of patients needing medical or surgical treatment for BOO related to BPH.

Finally, none of the 20 patients included in our series required further surgical interventions during the 6-month follow-up. Nonetheless, despite adequate patient selection and indication, the literature reports that 2–6% of PUL patients will require surgical re-intervention [19].

In the past few years, several authors have evaluated the role of MIST in the treatment of benign prostatic hyperplasia [5,20,21,22,23,24,25,26,27].

Numerous reports have demonstrated that PUL leads to significant improvements in urological outcomes, with reductions in IPSS from 39% to 52%, increases in Qmax from 32% to 59%, and enhancements in QoL from 48% to 53% [6,8,28,29,30,31,32]. A meta-analysis of both retrospective and prospective trials confirmed these overall improvements following PUL. A Cochrane review comparing PUL to TURP in sham RCTs and direct RCTs concluded that while PUL is less effective than TURP in improving IPSS and Qmax in both the short- and long-term, QoL outcomes may be similar between the two procedures. Additionally, a multicenter study with one-, three-, and five-year follow-ups of the treated cohort showed durable improvements in IPSS, QoL, and Qmax after PUL, with improvement rates of 36%, 50%, and 44% at the 60-month follow-up, respectively. A network meta-analysis, including outcomes data at three to six months follow-up, found that IPSS improvements were similarly high after both TURP and PUL. However, objective outcomes such as PVR (post-void residual) and Qmax showed the greatest improvement after TURP, with PUL offering less pronounced progress. In a retrospective observational study involving 1413 patients from North America and Australia, patients were divided into those still voiding (Group A) and those in retention (Group B). The results from Group A were comparable to clinical trials, and in Group B, 69% of the 165 patients were catheter-free after five days, 83% after one month, and 89% by the end of the study. Common post-operative complications included haematuria (16–63%), dysuria (25–58%), pelvic pain (5–17.9%), urgency (7.1–10%), transient incontinence (3.6–16%), and UTI (2.9–11%). Most of these symptoms were mild to moderate in severity and resolved within two to four weeks after the procedure. A RCT comparing PUL to TURP measured surgical recovery using a validated instrument and found that recovery from PUL was more rapid and extensive in the first three to six months. Additionally, a systematic review and meta-analysis found that sexual function, particularly erectile and ejaculatory function, remained stable or improved slightly during the 24-month follow-up.

Although their exact role is still a matter of debate, they might be an alternative to medical treatment before definitive treatment. The principal advantage of most of these techniques is the preservation of the ejaculatory function, which is of utmost importance for young men. In fact, several modifications of these procedures have been developed with mixed results [32,33,34,35,36]. Notable among these are enucleation transurethral resection of the prostate, enucleation photoselective vaporization of the prostate, ejaculatory hood sparing holmium laser enucleation of the prostate, and median lobe enucleation. Additionally, the Madigan simple prostatectomy (open, laparoscopic, and robotic) with a urethral sparing technique is currently under evaluation. Meanwhile, techniques such as interstitial laser coagulation (ILC), visual laser ablation of the prostate (VLAP), and transurethral microwave thermotherapy (TUMT) have become nearly obsolete.

Another important matter to discuss is the utility of urodynamic studies before and after surgical interventions for BPH. PFSs are considered the gold standard in the diagnosis of BOO in male patients. However, according to the EAU guidelines, PFSs should be performed only in selected patients before surgery or when evaluation of the underlying pathophysiology of LUTSs is warranted. Moreover, the clinical benefit for the patient is often lacking; the procedure is invasive and carries a consistent risk of complications (discomfort, hematuria, fever, urinary tract infection). Lately, efforts have been made to avoid PFSs through less-invasive methods. BOO diagnoses by bladder/detrusor wall thickness, bladder weight measures, and intraprostatic bladder protrusion have been proposed by many authors [37,38]. However, all these methods are still considered investigational. As well some authors have proposed nomograms to predict BOO by combining different non-invasive measures. These nomograms, although validated, present fair accuracies and cannot be considered an alternative to PFS. Finally, some authors proposed to use the PVR ratio to predict BOO and DU with conflicting results [39]. Given the available evidence and notwithstanding big efforts to avoid the use of PFS, PFS remains an important diagnostic tool to evaluate the pathophysiology of lower urinary tract dysfunctions.

In recent years, the advent of artificial intelligence (AI) and natural language processing (NLP) has marked a game-changing moment in urological research. Artificial intelligence (AI) is a branch of computer science focused on creating machines and systems capable of performing tasks that typically require human intelligence. In urology, and particularly in functional urology, AI is striving to overcome the limitations of current predictive models and has the potential to enhance personalized medicine through its ability to analyze large datasets. Additionally, the introduction of natural language processing (NLP) and tools like ChatGPT may assist patients in obtaining accurate information in the field of functional urology. For example, some studies have assessed the accuracy of NLP in responding to questions based on urological guidelines, yielding encouraging results [39,40].

The role of AI and NLP in managing patients with BOO due to BPH who are considering surgical treatment is still being defined [40,41,42,43]. AI has also been integrated into surgical procedures to assist surgeons during operations, with its use being particularly well-tested in robotic surgery. However, its application in endoscopic procedures remains largely anecdotal. For example, AI could potentially guide surgeons during the PUL procedure by helping to determine the optimal angle and pressure to apply, thereby reducing the risk of implant misplacement.

A significant limitation of the current literature on BPH surgery is the lack of well-defined outcomes [44,45,46,47,48]. Existing guidelines on non-neurogenic male LUTS do not provide a clear definition of a successful outcome following surgical treatment. Overall, surgical outcomes can be categorized into several areas: symptom improvement, flow improvement, absence of perioperative complications, recovery after surgery, impact on sexual function, absence of urinary incontinence, and patient-reported outcomes.

The literature highlights an important debate regarding the ideal proxy for evaluating surgical outcomes in BOO [44,45,46,47,48]. Many studies have assessed predictors of BOO surgical outcomes using varying definitions, but this inconsistency introduces significant bias when comparing different studies [44,45,46,47,48]. Recently, there has been a growing emphasis on patient-reported outcomes, which are not yet considered a standard proxy for BOO surgical outcomes in the available literature. It is likely that relying solely on Qmax or IPSS cut-offs does not accurately represent a successful outcome [44,45,46,47,48]. Thus, future research should focus on identifying predictors of patient satisfaction or regret to better determine which patients will truly benefit from surgery [44,45,46,47,48]. Alternatively, international guidelines should establish clear and specific criteria for defining a successful BOO surgery.

Additionally, it is crucial to differentiate outcomes between standard and minimally invasive treatments [21,49,50,51]. Patients and surgeons cannot expect identical outcomes and morbidity when comparing ablative and non-ablative treatments. Minimally invasive procedures, especially nonablative techniques such as PUL and iTind, have the advantage of sparing ejaculation, being an attractive alternative to medical treatment and surgery for young patients [21,49,50,51]. The results of these techniques cannot be compared to classic ablative techniques and probably represent a bridge between medical treatment and definitive surgical intervention. As for the time being, the number of minimally invasive techniques and classic techniques warrants careful patient selection in order to offer a tailored treatment to the patient [21,49,50,51]. A patient-centered approach is essential to ensure patient satisfaction and surgical outcomes. The analysis of individual patient data may provide insights into some of our study’s findings. Similar to the results reported by Muller et al., we did not find significant differences between preoperative and postoperative median pdetQmax [19]. Our results suggest that the urodynamic effect of the PUL procedure is moderate, as all parameters showed slight improvement, and not all patients experienced urodynamic benefits. Specifically, 16 out of 20 patients (80%) showed improvement in the Schäfer class, and 14 out of 20 patients (70%) exhibited improvements in pdetQmax. Although the study is not sufficiently powered to analyze predictors of lack of improvement in BOO parameters, patients with persistent obstruction had a prostate volume greater than 50 cc, which may partially explain their poor response to PUL. Overall, careful patient selection is crucial for identifying candidates who will benefit from PUL, and ongoing studies will soon help identify predictors of poor outcomes.

Although the study is not sufficiently powered to analyze predictors of lack of improvement in BOO parameters, patients with persistent obstruction had a prostate volume greater than 50 cc, which may partially explain their poor response to PUL [21,49,50,51]. Overall, careful patient selection is crucial for identifying candidates who will benefit from PUL, and ongoing studies will soon help identify predictors of poor outcomes. In recent years, various authors have investigated potential predictors of BOO, including non-invasive PFS, bladder wall thickness, prostate shape, intraprostatic protrusion, prostatic urethral angle, and transitional zone index [49,50,51]. Despite these efforts, none of the non-invasive parameters can currently replace PFS for diagnosing BOO [49,50,51]. However, particularly concerning the PUL procedure, future studies should evaluate the potential role of non-invasive BOO proxies as predictors of surgical outcomes after PUL. While some researchers suggest that PUL can be safely performed in patients with a certain degree of median lobe involvement [52], the procedure can be challenging, and outcomes may not always be satisfactory.

We have to acknowledge some limitations to our study. The small sample size was the main limitation of our study, although the study was designed to confirm the urodynamic effects of PUL. Moreover, our study applies to a cohort population of highly symptomatic patients with suboptimal detrusor contractility and may not be extended to other cohorts with different characteristics. Finally, the lack of long-term follow-up may be considered a limitation of our study. Nonetheless, a study is ongoing to evaluate the long-term urodynamic effects of PUL.

## 5. Conclusions

In our study, the PUL procedure improved both urodynamic parameters and QoL, thanks to symptom relief in male patients with LUTS caused by BPH. It also demonstrated short operation time, good safety, and a quick post-procedural recovery. IPSS was almost halved at 6 months follow-up, but these results cannot be fully explained by the modest UDN effect observed. This disparity could influence PUL efficacy, and its understanding could improve patients’ satisfaction and reduce the need for further interventions. Over the past decade, there has been a rapidly growing enthusiasm for AI and its applications across various urological subspecialties. However, the role of AI in BPH surgery remains underexplored. Integrating AI into areas such as diagnostics, treatment planning, and patient management could potentially enhance clinical care and deepen our pathophysiological understanding of functional urologic conditions. For instance, AI systems could analyze large datasets to recommend personalized treatment plans tailored to individual patient profiles, thereby optimizing surgical procedures, increasing precision, reducing the risk of complications, and ultimately improving outcomes.

## Figures and Tables

**Figure 1 diagnostics-14-02110-f001:**
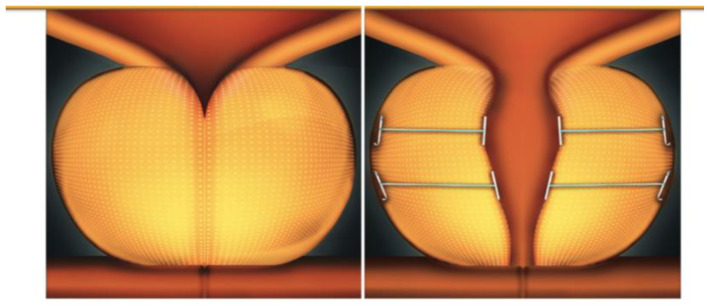
The Prostatic Urethral Lift procedure (Provided by Teleflex, Rome, Italy).

**Figure 2 diagnostics-14-02110-f002:**
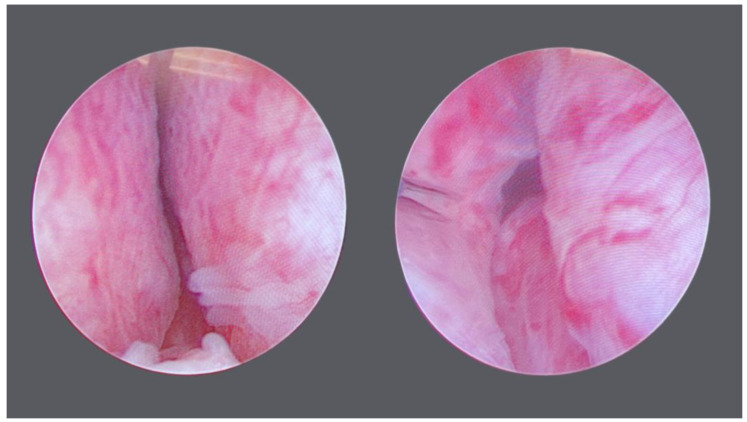
Endoscopic View before and after the procedure.

**Table 1 diagnostics-14-02110-t001:** Baseline Characteristics of the Cohort.

	Baseline
Age (years)	63 (56–69)
BMI (kg/m^2^)	26 (24–28)
Hypertension	10/20 (50%)
Diabetes	2/20 (10%)
Alpha-blockers treatment	16/20 (80%)
5-ARI treatment	6/20 (30%)
Antimuscarinic treatment	1/20 (5%)
Prostate Volume (cc)	43 (38–50)
PSA (ng/mL)	2.8 (1.5–3.8)
vIPSS	10 (7–13)
sIPSS	6.5 (4–8)
IPSS	16 (13–23)
MSHQ	10 (3–13)
IIEF	23(21–25)
Free flow Qmax (mL/s)	8.5 (7–11)

Data are presented as Median (Interquartile Range) and percentages xx/yy (%).

**Table 2 diagnostics-14-02110-t002:** Clinical Improvements after Prostatic Urethral Lift.

	Baseline	6 Months	*p*
vIPSS	9.5 (7–13)	5 (3–6)	0.001
sIPSS	6.5 (4–8)	4 (3–6)	0.001
IPSS	16 (13–23)	10.5 (7–12)	0.001
Free flow Qmax	8.5 (7–11)	11.5 (10–15.5)	0.001
IIEF	23 (21–25)	22 (21–25)	0.255
MSHQ	10 (3–13)	12.5 (6–14)	0.001

Data are presented as Median (Interquartile Range).

**Table 3 diagnostics-14-02110-t003:** Urodynamic improvements after Prostatic Urethral Lift.

	Baseline	6 Months	*p*
UDN-Qmax (mL/s)	8 (5–10)	11 (8–13)	0.001
pdetQmax (cmH20)	55 (48–66)	49 (32–61)	0.231
BCI	95 (78–110)	100 (88–127)	0.001
BOOI	39 (29–54)	18 (11–38)	0.001
Schäfer Class	III (II–IV)	II (I–II)	0.006

Data are presented as Median (Interquartile Range).

## Data Availability

Data will be available upon request.

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
