# Peer review of "Evaluation of Peri-Operative Outcomes after Prostatic Urethral Lift with Emphasis on Urodynamic Changes, Symptom Improvement and Sexual Function"

_diagnostics, 2024, doi:10.3390/diagnostics14192110_

Round 1

Reviewer 1 Report

Comments and Suggestions for Authors

This manuscript reported the urodynamic results in male patients with LUTS/BPH undergoing prostatic urethral lift (PUL) procedure. The case number is small and there are several critical points the authors should reply or revise:

1) The baseline voiding pressure (Pdet) is high and postoperative Pdet did not significantly decrease. It is possible that part of patients did not have bladder outlet obstruction at baseline. 

2) The treatment outcome was not as good as that in patients who underwent classical or laser TURP. The authors should comment on this point.

3) If the authors can divide patients into BOO and non-BOO subgroup and compare the treatment results between thee two groups, the results might be more valuable in selecting patients who intent to undergo PUL procedure.

4) Although the prostate volume cannot predict BOO or not, the authors might report the PUA or TZI, and compare the treatment outcome for subgroups with high and low PUA or TZI. 

Comments on the Quality of English Language

Authors should carefully checked spelling and format of the text. 

Author Response

This manuscript reported the urodynamic results in male patients with LUTS/BPH undergoing prostatic urethral lift (PUL) procedure. The case number is small and there are several critical points the authors should reply or revise:

1) The baseline voiding pressure (Pdet) is high and postoperative Pdet did not significantly decrease. It is possible that part of patients did not have bladder outlet obstruction at baseline. 

We thank the reviewer for his/her comment and for the possibility to improve our manuscript.  As per inclusion criteria all patients presented BOO defined as Shaffer class≥  2. We have reported in the manuscript detrusor pressure at maximum flow rate. Similarly, to Muller et al manuscript we did not found significant differences between pre and postoperative UDN pdetQmax. As suggested by our results the urodynamic effect of Urolift procedure is moderate considering that all the parameters improve slightly and not all patients present urodynamic improvements. The lack of statistically significant differences in PdetQmax parameter is explained by the sample size of our study and by the moderate effect of Urolift in terms of UDN improvement.  In order to strengthen our manuscript we have added a supplementary table with the urodynamic data of the 20 patients and  we have improved the discussion section.

Please see New Supplementary table attached

See Discussion section:

The analysis of individual patient data may provide insights into some of our study's findings. Similar to the results reported by Muller et al., we did not find significant differences between preoperative and postoperative median pdetQmax. Our results suggest that the urodynamic effect of the Urolift procedure is moderate, as all parameters showed slight improvement, and not all patients experienced urodynamic benefits. Specifically, 16 out of 20 patients (80%) showed improvement in Shaffer class, and 14 out of 20 patients (70%) exhibited improvements in pdetQmax. Although the study is not sufficiently powered to analyze predictors of lack of improvement in BOO parameters, patients with persistent obstruction had a prostate volume greater than 50 cc, which may partially explain their poor response to Urolift. Overall, careful patient selection is crucial for identifying candidates who will benefit from Urolift, and ongoing studies will soon help identify predictors of poor outcomes.

2) The treatment outcome was not as good as that in patients who underwent classical or laser TURP. The authors should comment on this point.

We thank the reviewer for his/her comment and for the possibility to improve our manuscript.  Minimally invasive procedures particularly non ablative techniques such as Urolift and iTind have the advantage of sparing ejaculation being an attractive alternative to medical treatment and surgery for young patients. Results of these techniques cannot be compared to classic ablative techniques and probably represent a bridge between medical treatment and definitive surgical intervention. As for the time being the number of minimally invasive techniques and classic techniques warrants a careful patient selection in order to offer a tailored the treatment to the patient. A patient centered approach is essential to ensure patient satisfaction. 

Discussion section has been improved as follows:

Additionally, it is crucial to differentiate outcomes between standard and minimally invasive treatments. Patients and surgeons cannot expect identical outcomes and morbidity when comparing ablative and non-ablative treatments. Minimally invasive procedures particularly non ablative techniques such as Urolift and iTind have the advantage of sparing ejaculation being an attractive alternative to medical treatment and surgery for young patients. Results of these techniques cannot be compared to classic ablative techniques and represent probably a bridge between medical treatment and definitive surgical intervention. As for the time being the number of minimally invasive techniques and classic techniques warrants a careful patient selection in order to offer a  tailored  treatment to the patient. A patient centered approach is essential to ensure patient satisfaction and surgical outcomes.

3) If the authors can divide patients into BOO and non-BOO subgroup and compare the treatment results between thee two groups, the results might be more valuable in selecting patients who intent to undergo PUL procedure.

We thank the reviewer for his/her comment and for the possibility to improve our manuscript.  As stated, earlier, patients with no BOO were excluded from the study.  This is now better specified in the methods section. Overall, 13/20 patients presented a Shaffer class ≥ 3 and 7 patients presented a Shaffer Class = 2. Study is not powered to compare these two groups however future studies should assess the impact of different pre-oparative parameters in predicting urolift outcome. Discussion section has been improved see next point.

4) Although the prostate volume cannot predict BOO or not, the authors might report the PUA or TZI, and compare the treatment outcome for subgroups with high and low PUA or TZI. 

We thank the reviewer for his/her comment. In our study all patients underwent suprapubic ultrasound to evaluate prostate volume as we do not routinely perform transrectal ultrasound before surgery. As suggested by the EAU guidelines prostate volume may be estimated with suprapubic ultrasound.  We agree with the reviewer that having PUA and TZI value may have added important information to our study however the objective of the study was not to evaluate non-invasive predictors of bladder outlet obstruction or surgical outcome. As for the time being none of the available non-invasive predictors for BOO has proven accurate and therefore are not recommended by the EAU guidelines. Although a previous study by our group demonstrated the role of TZI in predicting BOO, the narrow range of prostate volumes and the fact that all patients in the present study had BOO prevent a proper evaluation of TZI as a predictor of BOO. We have added a paragraph discussing this limitation to improve our manuscript.

See Discussion section:

Although the study is not sufficiently powered to analyse predictors of lack of improvement in BOO parameters, patients with persistent obstruction had a prostate volume greater than 50 cc, which may partially explain their poor response to Urolift. Overall, careful patient selection is crucial for identifying candidates who will benefit from Urolift, and ongoing studies will soon help identify predictors of poor outcomes. In recent years, various authors have investigated potential predictors of BOO, including non-invasive pressure flow studies PFS, bladder wall thickness, prostate shape, intraprostatic protrusion, prostatic urethral angle, and transitional zone index. Despite these efforts, none of the non-invasive parameters can currently replace PFS for diagnosing BOO. However, particularly concerning the UroLift procedure, future studies should evaluate the potential role of non-invasive BOO proxies as predictors of surgical outcomes after UroLift. While some researchers suggest that UroLift can be safely performed in patients with a certain degree of median lobe involvement, the procedure can be challenging, and outcomes may not always be satisfactory.

Reviewer 2 Report

Comments and Suggestions for Authors

Urodynamic effect of prostatic urethral lift

1.     The title is a bit confusing. You have not assessed the urodynamic effects of PUL, rather you have assessed the changes in UDS parameters after PUL. Also to include the secondary objectives, the title needs to be broader.  Example could be ‘study of peri-operative outcomes of PUL with emphasis on urodynamic changes’.

2.     Urolift is a trade name. better to avoid that as say prostatic urethral lift.

3.     Overall 20 patients studied but the improvement in Schafer class is available only for 18.

4.     Lot of spelling mistakes. Please get a spell-check done. Few examples:

Line 26: The spelling of Schafer is incorrect.

Line 35: prostatectomdepend: some mix-up of words

Line 42: discomfort spelt incorrectly

Line 43: ejaculation spelt incorrectly

Line 45: aquablation spelt incorrectly.

I am not pointing all errors but please check thoroughly.

5.     Line 46 you say that most of these are considered experimental. This is not true as a few of these have found place in the recent guidelines.

6.     The criticism of the Muller study in unjustified as when you are out to look for UDS results, why do you need to describe the standard UDS procedure and why should you look at the sexual function. These would be out of bounds of the study.

7.     Pressure-flow study is always transurethral (line 85)

8.     Schäfer Class ≥ 3 would diagnose BOO and not BPO (line 91).

9.     Midstream urinalysis and culture are performed before and not at the time of UDS assessment (line 96-97).

10.   Instead of cystoscopy tube, cystoscopy sheath is a better term.

11.   Can you use the photograph of the Urolift procedure provided by the Teleflex company.

12.   You have said, Enlargement was verified through cystoscopy anterior to the verumontanum. I think you mean opening of the anterior chamber of the prostatic urethra confirmed cystoscopically.

13.   Line 124: change adopted to needed

14.   Line 125: change time to discharge to duration of hospitalization; also when needed does not sound good.

15.    IN Table 1: for prostate volume, IPSS and other parameters, the range is not written correctly. Instead of 38-50, you have written 38/50. Make this changes in for all parameters. Similar change is needed in table no 2.

16: your discussion is very long and not focused to the present topic. please reduce it. 

Comments on the Quality of English Language

Provided in the comments to authors.

Lot of spelling and typographic errors

Author Response

Reviewer 2:

Urodynamic effect of prostatic urethral lift

We thank the reviewer for the extensive and useful revision of our manuscript..

  1. The title is a bit confusing. You have not assessed the urodynamic effects of PUL, rather you have assessed the changes in UDS parameters after PUL. Also to include the secondary objectives, the title needs to be broader.  Example could be ‘study of peri-operative outcomes of PUL with emphasis on urodynamic changes’.

We thank the reviewer for his her comment title has been changes as suggested by the reviewer.

See New Title:

Evaluation of peri-operative outcomes after prostatic urethral lift with emphasis on urodynamic changes, symptoms improvement and sexual function

  1. Urolift is a trade name. better to avoid that as say prostatic urethral lift.

We thank the reviewer for his/her comment. Urolift has been avoided.

  1. Overall 20 patients studied but the improvement in Schafer class is available only for 18. We thank the reviewer for his/her comment we apologize for the typo error. Abstract has been updated
  2. Lot of spelling mistakes. Please get a spell-check done. Few examples:

Line 26: The spelling of Schafer is incorrect.

Line 35: prostatectomdepend: some mix-up of words

Line 42: discomfort spelt incorrectly

Line 43: ejaculation spelt incorrectly

Line 45: aquablation spelt incorrectly.

I am not pointing all errors but please check thoroughly.

We thank the reviewer for his/her comment. We apologize for the typos. The text has been reviewed by a native English speaker.

  1. Line 46 you say that most of these are considered experimental. This is not true as a few of these have found place in the recent guidelines.

We thank the reviewer for his/her comment we have amended the phrase it now reads:

‘Some of these techniques are still to be considered investigational and cannot be compared in terms of functional outcomes to the classic interventions, although their invasiveness is more favorable.’

  1. The criticism of the Muller study in unjustified as when you are out to look for UDS results, why do you need to describe the standard UDS procedure and why should you look at the sexual function. These would be out of bounds of the study.

We thank the reviewer for his/her comment. We agree with the reviewers’ comments, and we have changed the phrase accordingly.

See New Phrase:

‘The study by Muller is the only study on the subject available in the literature, however it takes more than one study to prove a hypothesis.’

  1. Pressure-flow study is always transurethral (line 85)

We thank the reviewer for his/her comment. Some non-invasive PFS exist with external cuffs so we have added transurethral to be more precise.

  1. Schäfer Class ≥ 3 would diagnose BOO and not BPO (line 91).

We thank the reviewer for his/her comment. We agree with the reviewer the phrase has been modified. We apologize for the typo error.

  1. Midstream urinalysis and culture are performed before and not at the time of UDS assessment (line 96-97).

We thank the reviewer for his/her comment. The text has been updated accordingly.

See methods section:

‘ Midstream urinalysis, including urinary sediment and culture performed before UDS, yielded negative results at the time of the urodynamic evaluation.’

  1. Instead of cystoscopy tube, cystoscopy sheath is a better term.

We thank the reviewer for his/her comment. The text has been updated accordingly.

  1. Can you use the photograph of the PUL procedure provided by the Teleflex company.

We thank the reviewer for his/her comment and for the suggestion. We have asked the permission to Teleflex to use the photograph and Teleflex has provided a new similar picture. See new picture. 

It is now specified on the image.

  1. You have said, Enlargement was verified through cystoscopy anterior to the verumontanum. I think you mean opening of the anterior chamber of the prostatic urethra confirmed cystoscopically.

We thank the reviewer for his/her comment the text has been amended according to his/her suggestions.

See Methods section:

Mean opening of the anterior chamber of the prostatic urethra was confirmed cystoscopically.

  1. Line 124: change adopted to needed

We thank the reviewer for his/her comment the text has been amended.

  1. Line 125: change time to discharge to duration of hospitalization; also when needed does not sound good.

We thank the reviewer for his/her comment the text has been amended as follows:

See Methods section:

In the postoperative setting, duration of hospitalization and catheterization time were evaluated.

  1. IN Table 1: for prostate volume, IPSS and other parameters, the range is not written correctly. Instead of 38-50, you have written 38/50. Make this changes in for all parameters. Similar change is needed in table no 2.

We thank the reviewer for his/her comment tables have been updated and we have better specified that data is presented as Median (Interquartile Range).

16: your discussion is very long and not focused to the present topic. please reduce it. 

We thank the reviewer for his/her comment. Discussion section has been improved and some sections deleted as suggested by the other reviewers. However as per editorial indications the manuscript must be at least 4 000 words.

Reviewer 3 Report

Comments and Suggestions for Authors

PFS was performed on 20 BPH patients before and 6 months after surgery to determine the effect of PUL on urodynamics. Although this is a study of a very small number of cases, it is considered a useful paper because such reports are few. However, I have the following concerns regarding the analysis of PFS data.

In Table 2, there is a decrease in BOOI while PdetQmax is unchanged; Qmax is significantly elevated for that reason, but why did Qmax increase in the absence of obstruction reduction? It is difficult to believe that the detrusor contractility improves at 6 months. Abdominal pressure urination may induce artificial Qmax increase and result in increased BCI and decreased BOOI.

To dispel these concerns, the authors can show the Uroflow waveforms before and after surgery in each of the 20 cases.

Other comments

1) Line 105 says general anesthesia and line 117 says spinal anesthesia, which is true?

2) In Table 1, AM treatment, what does it mean, antimuscarinic?

3) What do the (  ) in Table 1 and 2 mean? Range?

4) Discussion and conclusion are redundant. At least the description of AI has nothing to do with this paper.

Author Response

PFS was performed on 20 BPH patients before and 6 months after surgery to determine the effect of PUL on urodynamics. Although this is a study of a very small number of cases, it is considered a useful paper because such reports are few. However, I have the following concerns regarding the analysis of PFS data.
In Table 2, there is a decrease in BOOI while PdetQmax is unchanged; Qmax is significantly elevated for that reason, but why did Qmax increase in the absence of obstruction reduction? It is difficult to believe that the detrusor contractility improves at 6 months. Abdominal pressure urination may induce artificial Qmax increase and result in increased BCI and decreased BOOI.

To dispel these concerns, the authors can show the Uroflow waveforms before and after surgery in each of the 20 cases.

We thank the reviewer for his/her comment. We have reported in the manuscript detrusor pressure at maximum flow rate. Similarly, to Muller et al manuscript we did not find significant differences between pre and postoperative UDN. As suggested by our results the urodynamic effect of PUL procedure is moderate considering that all the parameters improve slightly and not all patients presented urodynamic improvements. The lack of statistically significant differences in PdetQmax parameter is explained by the sample size of our study and by the moderate effect of PUL in terms of UDN improvement. We agree with the reviewer that having single patient data may be useful. We cannot upload waveforms before and after surgery for privacy reasons. However, we have added a table with all the pre and post uroflow and urodynamic parameters of the 20 patients as supplementary material.  Moreover, a paragraph has been added to the discussion section.

Other comments

1) Line 105 says general anesthesia and line 117 says spinal anesthesia, which is true?

We apologize for the typo error. All patients were treated with spinal anaesthesia.

2) In Table 1, AM treatment, what does it mean, antimuscarinic?

AM is antimuscarinic. We have updated the table.

3) What do the (  ) in Table 1 and 2 mean? Range?

The () is interquartile range. It is now specified in methods section.

4) Discussion and conclusion are redundant. At least the description of AI has nothing to do with this paper.

We thank the reviewer for his/her suggestion.

The manuscript has been carefully reviewed and discussion has been improved. The AI part has been shortened referring only to its possible application to improve the PUL procedure.  

Round 2

Reviewer 1 Report

Comments and Suggestions for Authors

This manuscript has been adequately revised.

Comments on the Quality of English Language

This manuscript has been adequately revised.

Author Response

We thank the reviewer for his/her comments. 

The manuscript has been carefully checked for consistency, grammar and typos. 

Reviewer 3 Report

Comments and Suggestions for Authors

The same as above.

Thank the authors for revision of the manuscript according to the comments of the reviewer.  
Abstract must be completely revised because the content is different from the main text. PdetQmax was improved in abstract but not in main text.
For example, 14/20 presented an improvement in Schaffer class in abstract, but in result 12/20, in discussion 18/20.  Which is true?  age 66 in abstract, age 63 in maintext and table 1, which is correct? Qmax 10.4→13.9 in abstract, but in Table 2 8.5→11.5, which is correct?
In main text, sIPSS decreased by 4.5 and vIPSS by 2.5 points, but in Table 2 vPPS 10→5, sIPSS 6.5→4, Is it true?
There are many other conflicting numbers besides this one.
The authors should carefully check the consistency.

Author Response

Thank the authors for revision of the manuscript according to the comments of the reviewer.  

Abstract must be completely revised because the content is different from the main text.

We thank the reviewer for reviewing our manuscript and for the possibility to improve the text.

The mm section has been carefully reviewed and harmonized to the abstract.

PdetQmax was improved in abstract but not in main text.

We thank the reviewer for his/her comment. PdetQmax is now consistent in the abstract and in the main text.

For example, 14/20 presented an improvement in Schaffer class in abstract, but in result 12/20, in discussion 18/20.  Which is true? 

We thank the reviewer for the careful analysis of our manuscript consistency and we apologize for any typo. The manuscript has been carefully reviewed by our statistician. For instance definitive numbers are the following:

Improvement in Schaffer class in 16/20 patients

BOOI< 20 in 12/20 patients

Age 66 in abstract, age 63 in main text and table 1, which is correct?

We thank the reviewer for his/her comment. Initially we used means in the abstract however in order to maintain consistency and for the sake of clarity results have been harmonized. 

The correct median age is 63 we apologize for the typo error.

Qmax 10.4→13.9 in abstract, but in Table 2 8.5→11.5, which is correct?

We thank the reviewer for his/her comment. Initially we used means in the abstract however in order to maintain consistency and for the sake of clarity results have been harmonized.  

In main text, sIPSS decreased by 4.5 and vIPSS by 2.5 points, but in Table 2 vPPS 10→5, sIPSS 6.5→4, Is it true?

We thank the reviewer for his/her comment. We apologize for the typo error sIPSS and vIPSS were inverted in the text. As well we have avoided rounding of the vIPSS in Table 2 which now reads 9,5.

Results section and table shave been carefully reviewed by the statistician.  

There are many other conflicting numbers besides this one.
The authors should carefully check the consistency.

We thank the reviewer for his/her comment the manuscript has been completely reviewed by our statistician in order to avoid inconsistency.

Round 3

Reviewer 3 Report

Comments and Suggestions for Authors

The appropriate corrections were given.